# The economics of physical activity in low-income and middle-income countries: protocol for a systematic review

Priyanga Diloshini Ranasinghe,[1,2] Subhash Pokhrel,[3] Nana Kwame Anokye[3]

[1]Department of Health, Ministry of Health, Nutrition and Indigenous Medicine, Colombo, Sri Lanka
[2]Division of Health Sciences, Department of Clinical Sciences, Collage of Health and Life Sciences, Brunel University, London, UK
[3]Health Economics Theme, Institute of Environment, Health and Societies, Brunel University, London, UK

**Correspondence to**
Dr Priyanga Diloshini Ranasinghe;
priyangaran@yahoo.com, Priyanga.Ranasinghe@brunel.ac.uk

## ABSTRACT

**Introduction** Evidence on the economic costs of physical inactivity and the cost-effectiveness of physical activity interventions in low-income and middle-income countries (LMICs) is sparse, and fragmented where they are available. This is the first review aimed to summarise available evidence on economics of physical activity in LMICs, identify potential target variables for policy, and identify and report gaps in the current knowledge on economics of physical activity in LMICs.

**Methods and analysis** Peer-reviewed journal articles of observational, experimental, quasi-experimental and mixed-method studies on economics of physical activity in LMICs will be identified by a search of electronic databases; Scopus, Web of Science and SPORTDiscus. Websites of WHO, the National Institute for Health and Care Excellence international, World Bank and reference lists of included studies will be searched for relevant studies. The study selection process will be a two-stage approach; title and abstract screen for inclusion, followed by a review of selected full-text articles by two independent reviewers. Disagreements will be resolved by consensus and discussion with a third reviewer. Data will be extracted using standardised piloted data extraction forms. Risk of bias will be critically appraised using standard checklists based on study designs. Descriptive synthesis of data is planned. Where relevant, summaries of studies will be classified according to type of economic analysis, country or country category, population, intervention, comparator, outcome and study design. Meta-analysis will be performed where appropriate. This protocol for systematic review is prepared according to the Preferred Reporting Items for Systematic review and Meta-analysis for Protocols −2015 statement.

**Ethics and dissemination** Ethical approval is not obtained as original data will not be collected as part of this review. The completed review will be submitted for publication in a peer-reviewed journal and presented at conferences.

**PROSPERO registration number** CRD42018099856.

## Strengths and limitations of this study

► This is the first synthesis of literature on economics of physical activity in low-income and middle-income countries (LMICs).
► This study provides evidence-based recommendations for economic research practice on physical activity.
► This review presents evidence on the business case for physical activity interventions for decision making by policy makers in LMIC settings.
► This review could suffer from publication bias as it excludes studies that are not peer reviewed.
► This review focuses on studies written in the English language and could miss out on relevant literature published in other languages.

of the global health burden comprises NCDs which account for 1.4 billion disability-adjusted life years (DALYs).[3] At present most of the global NCD burden is from low-income and middle-income countries (LMICs) (as defined by[4] the World Bank, Classification 2017)[4] and this burden is largely attributable to levels of physical inactivity in the population.[5 6] Prevalence of physical inactivity in 2010 in upper middle-income countries was 25.4% (19.1%–33.7%), followed by 16.8% (11.8%–26.4%)%) in lower middle-income countries and 16.6% (11.8%–26.4%)%) in low-income countries showing that large populations in LMICs[7] are affected by physical inactivity and are therefore at risk of developing NCDs. The data available for the year 2004 show that 3.2 million deaths worldwide were due to physical inactivity of which 2.6 million were reported from LMICs.[8] LMICs have experienced urbanisation, shift of occupations from agriculture to industrial practice and modernisation including automation, and this has resulted in increased levels of physical inactivity in these populations.[9]

Increasing physical activity and reducing the burden of NCDs is the main goal of public health policies in LMICs.[10] It is a personal, corporate and government responsibility to

## INTRODUCTION
### Rationale

Physical inactivity is a global health challenge, the fourth leading cause of mortality worldwide[1] and the key risk factor for non-communicable diseases (NCDs) such as cardiovascular diseases, cancer and diabetes.[2] Almost a third

reduce the economic burden of unhealthy lifestyles in LMICs.[10] Economic analysis could uncover subtle characteristics of individual decision making on lifestyle choices, estimate the opportunity costs of not doing anything to improve population levels of physical activity and help evaluate what interventions work efficiently to achieve the goal of addressing the physical inactivity pandemic, particularly in LMICs.[11] The current focus of promoting physical activity has shifted from the traditional education approach to environment and policy approaches. Economic perspectives and public health perspectives complement each other in promotion of physical activity.[12] Economic perspectives inform the design of effective physical activity interventions (identifying economic barriers to physical activity) and the efficient allocation of resources—critical to the economies of LMICs.[12]

Our scoping exercise did not identify any published reviews that address the economics of physical activity in LMICs. Thus economic evidence on physical activity in LMICs could be limited or scattered.[13] Therefore, we designed this review to address the following research questions: (1) What is the available evidence base and research gaps on the economics of physical activity in LMICs? (2) What is the focus of and methods for underpinning economic research on physical activity in LMICs? (3) What are the target variables and cost-effective interventions for physical activity policy in LMICs?

This review will add to the scientific literature, provide an overview of the economic evidence base of physical activity in LMICs and fill the gaps in the available evidence regarding this. Providing an up-to-date synthesis of the economic evidence base is an efficient way of highlighting current research practices and new findings to inform researchers, and formulation of cost-effective physical activity programmes and policies.

### Objectives
The review will:
- Summarise available evidence on economics of physical activity in LMICs.
- Describe the focus and methods underpinning research on economics of physical activity in LMICs.
- Identify potential target variables and cost-effective interventions for physical activity in LMICs for policy formation.
- Identify and report gaps in research on economics of physical activity in LMICs to provide recommendations for the economic research agenda in LMICs.

### METHODS
The methods for this review were informed by previous reviews of economic analyses of physical activity[10 13–19] and are in line with recommendations on review of economic evidence.[20–22] This will be based on the 'Preferred Reporting Items for Systematic Review and Meta Analyses Protocols (PRISMA-P)' Statement[23] (online supplementary file 1).

### Eligibility criteria
Economic studies on physical activity in LMICs will be included. Specifically, the following study types will be included: (1) Economic evaluations of physical activity interventions. (2) Economic burden of physical inactivity. (3) Cost of physical activity participation. (4) Demand for physical activity. (5) Economic correlates of physical (in) activity. Physical activity is defined as any bodily movement produced by skeletal muscles that require energy expenditure—including activities undertaken while walking, playing, carrying out household chores, travelling and engaging in recreational pursuits.[24] Physical inactivity is considered as a lack of physical activity.[25]

Physical activity interventions will not be specifically defined for this review in order to identify any physical activity intervention in LMICs. However, studies of which an economic analysis was carried out along with the intervention will be included in this review. This strategy will enable consolidation of all available evidence on economics of physical activity related to physical activity interventions.

Eligibility criteria are determined by relevant elements of the population, intervention, comparator, outcome, study design (PICOS criteria).[26] Interventions and comparators will be applicable only to intervention studies.

### Inclusion criteria
- Study setting: Any setting of LMICs in accordance with the definition of LMIC by World Bank, Classification 2017.[4]
- Population: Any population in any age group across the life course.
- Intervention: Any physical activity intervention in which the economic evaluation of the intervention has been carried out.
- Comparator: Normal routine, no intervention.
- Outcomes: (1) Cost-effectiveness ratio, quality-adjusted life years (QUALY), incremental cost-effectiveness ratio assessed as the outcomes of physical activity interventions. (2) Cost of physical (in) activity in terms of healthcare cost and/or productivity loss and/or total cost of physical inactivity. (3) Measures of association of any economic variable with physical activity are defined as the primary outcomes of this review.
- Study design: Observational studies (cohort, case control, cross-sectional); correlational studies, experimental studies including randomised controlled trials; quasi-experimental studies; natural experiments; and economic evaluation studies.
- Studies reported only in the English language.

### Exclusion criteria
- Case reports, case series, letters to the editor, editorials, reviews, qualitative studies, unpublished theses, conference abstracts and any unobtainable texts.
- Studies published in a language other than English.

## Search strategy

The selection of databases was performed by a scoping review of methods of systematic reviews in the field.[13 17 18 27 28] We will search the following electronic databases to identify studies: Scopus (covers 100% MEDLINE coverage, 100% of EMBASE coverage and 100% of Compendex coverage[29]), Web of Science and SPORTDiscus. Websites of WHO, National Institute for Health and Care Excellence (NICE) international and World Bank will be searched for relevant studies. The reference lists of included studies will be searched for any relevant articles. Searches will include publications up to December 2017.

The search strategy was developed based on the scoping review that covered relevant reviews on economic studies on physical activity,[10 13–19] reviews on physical activity[27 30 31] and reviews on economic evaluations.[28 32 33] The draft search strategy was then reviewed by subject experts and a subject liaison librarian to optimise the sensitivity and specificity of the search. Online supplementary file 2 shows a sample search strategy.

## Study selection

A two-stage approach involving three independent reviewers will be used to select relevant papers.

At the first stage, two reviewers (PDR and NKA) will independently screen the titles and abstracts of identified papers. Disagreement will be resolved through discussions with a third reviewer (SP). We will include a study if in doubt about its inclusion. For example, if the country setting of a study is unclear in the title and abstract, we will take it forward to the next stage.

At the second stage, the full text of papers selected from stage 1, will be reviewed independently by two reviewers (PDR and NKA). Selected and excluded papers, with reasons, will be discussed by the two reviewers and a third reviewer (SP) at a consensus meeting and disagreement will be resolved by real time consensus. Corresponding authors will be contacted via email for clarification if needed. The study selection process will be presented in a PRISMA flow chart[34] along with the reasons for the exclusion of studies.

## Data management and extraction

Endnote X7 software will be used to manage the search results. Data extraction will be performed using a standardised pilot-tested data extraction form developed based on relevant data extraction forms from relevant reviews.[17 35 36] The adaptation process involved matching the review objectives with the available data extraction forms. Data from the final selected full-text articles will be extracted by one reviewer (PDR). To ensure quality of data extraction, a second reviewer (NKA) will independently extract data from a random selection of 50% of the final articles. Disagreements between the two reviewers will be discussed and resolved through discussions with the third reviewer (SP). Any unresolved disagreements will be reported in the final report. Online supplementary file 3 shows the draft data extraction form. Data will be extracted for the following items: general information, characteristics of the study, characteristics of the population/condition/intervention, data sources/data analysis/ outcomes, conclusions and the way forward suggested by the authors, challenges and quality assessment.

## Risk of bias and quality assessment

Risk of bias will be assessed by a one reviewer (PDR) for all the selected articles using standard checklists based on the study design. A second reviewer (NKA) will also independently assess the risk of bias on 50% of the randomly selected articles included.

Checklists are considered a reliable means of ensuring that all studies included are critically appraised in a standard way.[26] These will assess the information bias which could occur due to the methods, study design, data collection, data analysis and interpretation, and selection bias, where relevant. The most appropriate checklists for each expected study design for risk of bias assessment were selected based on relevant literature in this area[15 17 36–40] and by consensus with all three reviewers (online supplementary file 4). Risk of bias will be assessed using the following tools:

► Economic evaluation studies
  The Drummond checklist[41] which consists of 35 items will be used. It mainly assesses the quality of the study in relation to methods, study design, data collection, data analysis and interpretations. If an economic evaluation study is based on a decision analytical model, the Phillips[42] checklist will be used in addition.
► Cost of illness studies
  The Larg & Moss checklist[43] will be used to assess the quality of the study. It would assess the possible biases related to the analytical framework: what costs should have been measured, methodology and data; how well were resources used and productivity losses measured including representativeness of data for the study population, analysis and reporting. Both internal validity and external validity will be appraised.
► Studies on correlation or association
  The checklist recommended by NICE[44] will be used. Risk of bias related to sample selection, method of selection of exposure/comparison group, outcomes and analysis will be assessed. Both internal and external validity will be appraised.

Quality grading will be given to each included study based on the overall quality of the study. Grading for internal validity and external validity will be reported separately, where relevant. The grading process will follow the NICE recommendations[37] (figure 1):

► Good quality (++): almost all checklist criteria have been fulfilled; where they have not been fulfilled the conclusions are very unlikely to alter.
► Moderate quality (+): some of the checklist criteria are fulfilled; where they have not been fulfilled or not adequately described, conclusions are unlikely to alter.

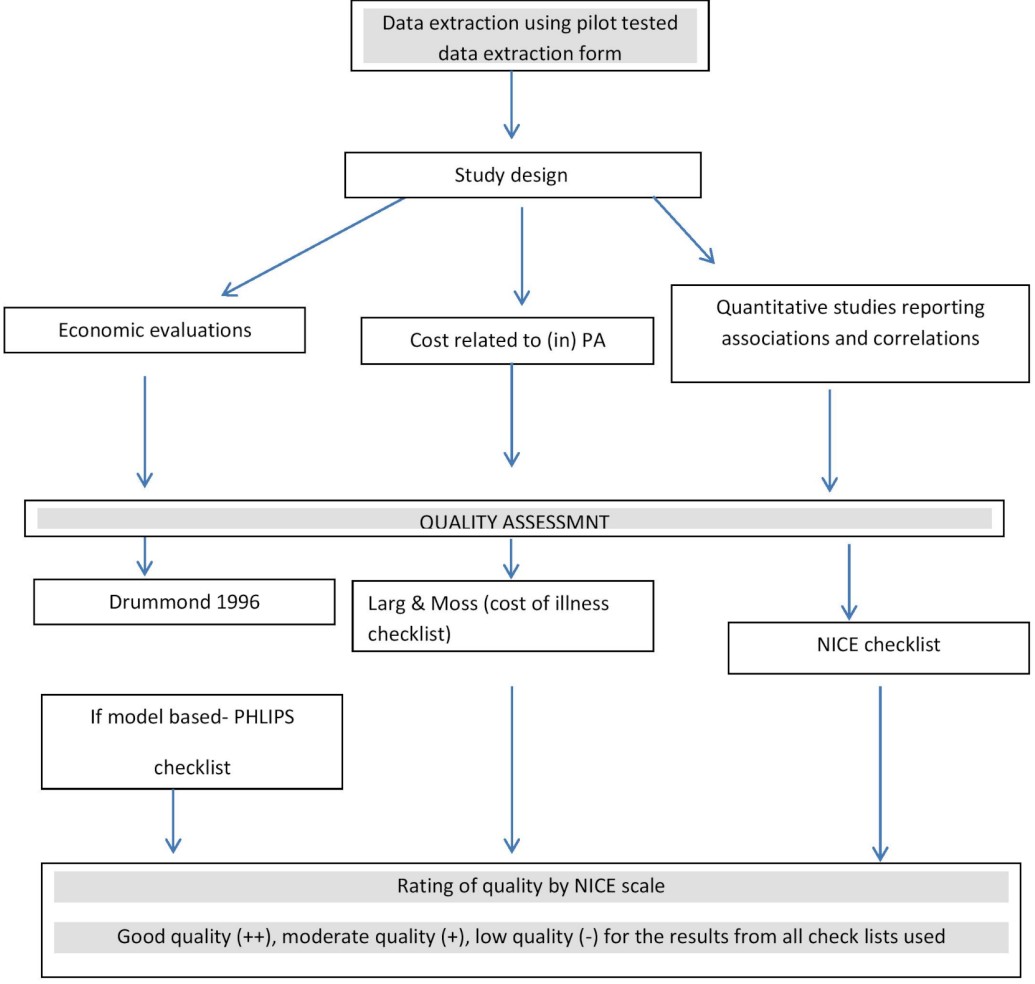

**Figure 1** Economics of physical activity (PA) in low-income and middle-income countries (LMICs)—plan for data extraction and the quality assessment process. NICE, National Institute for Health and Care Excellence.

► Low quality (−): few or no checklist criteria are fulfilled, and conclusions are likely or very likely to alter.

► The quality of the included studies will be reported and critically appraised for each study. However, studies will not be excluded based on quality, as this review is particularly aimed to capture the evidence base on economics of physical activity in LMICs and appraise the methods underpinning the evidence base.

### Data synthesis and reporting

A descriptive synthesis of data is planned due to the expected heterogeneity of the studies. Summaries of studies will be classified according to type of economic analysis, country or country category, characteristics of the study by PICOS criteria (where applicable); population, intervention, comparator, outcome, study design and quality of studies to describe the overview of available economic evidence base of physical activity in LMICs. Where the numbers permit, summaries of the studies will be categorised based on the focus of the study; economic burden of physical inactivity, economic analysis of physical activity interventions, economic

correlates of physical activity including association of physical activity and cost (healthcare cost and/or productivity loss). Where appropriate, we will conduct a meta-analysis to quantify the effect sizes. Such analyses will adjust for between-study heterogeneity using random-effects models. The I-squared ($I^2$) statistic will be used to evaluate the heterogeneity related to the findings of the papers. Quantitative analysis will be conducted using STATA V.13 software.

We will report the number of studies where data extraction items were not applicable, as an indicator of the quality of reporting. Author-stated limitations of included studies and recommendations for future research and policy will be presented. This review will be reported in accordance with the PRISMA 2009 statement.[34] The study selection process will be illustrated by a flow diagram that will include the reasons for exclusion of studies at each stage. The search strategy, data extraction forms and quality assessment tools will be published as online supplementary files. This protocol will not be amended and any changes will be described and discussed in the final report.

## Study status

A scoping review of economics of physical activity in LMICs was carried out in November 2017 to inform the methods of the current review. It identified systematic reviews conducted on economics of physical activity and the gaps in knowledge. It has not been submitted for publication. A review of economics of physical activity in LMICs was designed and methods were developed in December 2017. The search strategy was piloted in January 2018 and the protocol was developed in February 2018. The first submission of the protocol for peer review was on 1 March 2018. Data collection will be commenced upon approval of the protocol for publication and will be completed in 2 months.

## Patient and public involvement

Patients and public are not directly involved in this study as original data will not be collected.

Best practice guidelines to conduct systematic reviews[33] will be followed.

## DISCUSSION

The level of physical inactivity of the population in LMICs is likely to increase giving rise to a multitude of public health and economic consequences, including rising healthcare costs due to increased NCD-related treatments, increased DALYs and productivity losses. Economic research on physical activity from the perspective of LMICs should therefore be considered a priority. This is because increased physical inactivity leads to uncontrolled healthcare needs in LMICs, where resources (including that for healthcare) are limited and often shrinking in real terms. Pooling of the available research evidence on economics of physical activity in LMICs will thus reveal the current knowledge on economics of physical activity in LMICs, thereby guiding the future of national, regional and local policies around physical activity.

This will be the first study to review the evidence base on economics of physical activity in the context of LMICs which can be considered as the main strength of this study. This will identify gaps in knowledge to support future studies in this area. However, we will only include those studies that are published as peer-reviewed articles and are published in the English language. Thus, possibility of publication bias and language bias cannot be excluded. Furthermore, by assessing and reporting the quality of included studies using standard quality assessment checklists, the possibility of reporting bias of this review can be minimised.

WHO is in the process of drafting the 'Global action plan on physical activity (GAPPA) to be implemented from 2018 to 2030'.[45] This is a timely study as the evidence from this research will provide useful information for implementation and prioritise the actions of GAPPA in LMICs. Evidence from this research will be useful for policy makers and stakeholders dealing with physical activity promotion at individual, local, national, regional and global levels, with special reference to LMICs. Further, this review will be an important base for a research agenda on economics of physical activity in LMICs. This will be a useful guide for researchers to design research on economics and physical activity with sound methodology, based on the research needs of LMICs. Furthermore, information gathered from this research will guide funding agencies for effective allocation of resources.

**Contributors** SP and NKA developed the idea for the review with inputs from PDR. PDR wrote the first draft. SP and NKA revised the protocol. NKA will act as guarantor of the review.

**Funding** The authors have not declared a specific grant for this research from any funding agency in the public, commercial or not-for-profit sectors.

**Competing interests** None declared.

**Patient consent for publication** Not required.

**Provenance and peer review** Not commissioned; externally peer reviewed.

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
