## [Reviewer comments · BMJ Open]

ARTICLE DETAILS

TITLE (PROVISIONAL)	The economics of physical activity in low and middle income countries: protocol for a systematic review
AUTHORS	Ranasinghe, Priyanga; Pokhrel, Subhash; Anokye, Nana

VERSION 1 – REVIEW

REVIEWER	José Luis Márquez Andrade Universidad de Santiago de Chile, Chile
REVIEW RETURNED	24-Mar-2018

GENERAL COMMENTS	With respect to the manuscript by Dr. Ranasinghe et al., I must point out that it describes a protocol for a systematic review about the economics of physical activity in low and middle-income countries. The article addresses a current topic of great interest. The structure is appropriate and the language is clear. In my opinion, what needs attention is the following: 1.-The abstract is clear although some information may be added to improve the description of review (see below).2.-In the article's summary it is convenient to define if the authors consider that not including publications that are not peer-reviewed is a strength or a limitation. In the same way, excluding studies published in languages other than English should be declared as a limitation, since it may imply a language bias.3.-The Introduction is properly written and provides an adequate justification for the importance of the review. However, there is weakness in prioritizing the objectives, perhaps due to the lack of a clear and explicit review question.4.-Although the outcomes of selected articles do not seem to be an exclusion criteria, to detail some examples could help to clarify the objectives of the protocol.5.-In the data management and extraction section, the authors indicate that, in order to ensure the quality of data extraction, a second reviewer will repeat the action in a random selection of articles. It would be advisable to indicate the proportion of studies that will be evaluated doubly. The same is applicable to the risk of bias and quality assessment section. In this last section, on line 34 on page 8, there is a writing error ("all most all check list criteria...").6.-On line 38, page 10, it is necessary to write with initial capital letters "world health organization".7.-In figure 1, the authors must correct Larg's last name.8.-The supplementary material seems appropriate and pertinent, however, with respect to file 4, the table is difficult to understand and the rationale for the choice of tools presented in the conclusions is based on its frequency of use rather than its relation to the objective of evaluation. I think that authors should improve the rationale for the selection of quality assessment tools.9.-Finally, in the References section there are multiple errors and it is necessary that the authors review in detail the citation format
---

	required by the journal. Just to give some examples, it is necessary to indicate the date of access to the reviewed websites and complete the reference 15.
--	---

REVIEWER	Tuulikki Sjögren Faculty of Sport and Health Sciences, University of Jyväskylä, Finland
REVIEW RETURNED	21-May-2018

GENERAL COMMENTS	Research topic, the cost-effectiveness of physical activity, is very important worldwide. However, the manuscript of the study should be reinforced. Below are listed the items to be specified:  1) Physical activity and physical activity interventions should be defined more specifically. 2) This physical activity definition also related to the definition of outcomes results, which should be further defined (primary and secondary outcomes) 3) Defined more specifically, what new your research will bring? How the earlier studies results can be considered as limited and scattered? (abstract and p. 4) 4) In abstract should list all databases to be used (n=5) 5) A descriptive synthesis should describe more specifically, what characteristics? Take advantage of PICOS criteria 6) Take advantage of PICOS criteria also when define inclusion and exclusion criteria of the study. 7) Define more precisely in the text (Methods) about the risks you are talking about (p. 8) and how you will be control them. 8) Define more precisely in the text (Discussion) about the risks you are talking about (p. 10 and) how you will be control them.
--

VERSION 1 – AUTHOR RESPONSE

Reviewer 1 comments	What did we do	Where the changes are made
6. The abstract is clear although some information may be added to improve the description of review (see below).	Abstract improved	P2
7. In the article's summary it is convenient to define if the authors consider that not including publications that are not peer-reviewed is a strength or a limitation. In the same way, excluding studies published in languages other than English should be declared as a limitation, since it may imply a language bias.	Done	P3,L5=56
8. The Introduction is properly written and provides an adequate justification for the importance of the review. However, there is weakness in prioritizing the objectives, perhaps due to the lack of a clear and explicit review question.	The objectives restructured in priority order. The research questions included.	P5,L33-53 P5,L12-18
9. Although the outcomes of selected articles do not seem to be an exclusion criteria, to detail some examples could help to clarify the objectives of the protocol.	Expected outcomes are detailed as the fifth inclusion criteria.	P7,L11-20
10. In the data management and extraction	Percentage of articles evaluated doubly	P9,L9

section, the authors indicate that, in order to ensure the quality of data extraction, a second reviewer will repeat the action in a random selection of articles. It would be advisable to indicate the proportion of studies that will be evaluated doubly. The same is applicable to the risk of bias and quality assessment section. In this last section, on line 34 on page 8, there is a writing error (“all most all check list criteria...”).	indicated under data extraction and quality assessment section. Done	and L35 P10 L41
11. On line 38, page 10, it is necessary to write with initial capital letters “world health organization”.	Corrected as WHO	P13, L27
12. In figure 1, the authors must correct Larg's last name.	Done	figure 1 revised
13. The supplementary material seems appropriate and pertinent, however, with respect to file 4, the table is difficult to understand and the rationale for the choice of tools presented in the conclusions is based on its frequency of use rather than its relation to the objective of evaluation. I think that authors should improve the rationale for the selection of quality assessment tools.	Table on supplementary file 4 modified. Rationale for selection of quality assessment checklists explained in relation to objective of evaluation in the revised manuscript.	supplementary file 4 revised P9, L37-56, P10,L3-31
14. Finally, in the References section there are multiple errors and it is necessary that the authors review in detail the citation format required by the journal. Just to give some examples, it is necessary to indicate the date of access to the reviewed websites and complete the reference 15.	Done.	P14-22

Reviewer 2 Comments	What we do	Where the changes are made
15. Physical activity and physical activity interventions should be defined more specifically.	Physical activity defined more specifically. Physical activity interventions will not be specifically defined for this review in order to identify any physical activity intervention carried out in LMIC, along with an economic analysis.	P6, L30-34 P6, L37-43
16. This physical activity definition also related to the definition of outcomes results, which should be further defined (primary and secondary outcomes)	Done	P7,L 11-20

17. Defined more specifically, what new your research will bring? How the earlier studies results can be considered as limited and scattered? (abstract and p. 4)	Done.	P2 L6-12 P5, L8-12 P5, L21-30
18. In abstract should list all databases to be used (n=5)	All 3 data bases to be used in the search include in the abstract; Scopus, SPORTDiscus. Websites and reference lists to be search added to the abstract. Some sentences are shortened with tract changes to keep the word count under 300 with new additions.	P2, L23-28 Abstract
19. A descriptive synthesis should describe more specifically, what characteristics? Take advantage of PICOS criteria	Data synthesis further described as suggested	P11 L13-28
20. Take advantage of PICOS criteria also when define inclusion and exclusion criteria of the study.	Done	P7 inclusion criteria number 2--6
21. Define more precisely in the text (Methods) about the risks you are talking about (p. 8) and how you will be control them.	Possible risk of biases and plan to control them is described as suggested.	P10
22. Define more precisely in the text (Discussion) about the risks you are talking about (p. 10 and) how you will be control them.	Discussion is further improved as suggested	P13 L15-25

VERSION 2 – REVIEW

REVIEWER	José Luis Márquez Andrade Universidad de Santiago de Chile
REVIEW RETURNED	23-Jul-2018
GENERAL COMMENTS	In this version of the work by Ranasinghe et al., the authors adequately incorporated the suggestions made previously. As a final observation, I consider that including the current definition of physical inactivity, together with the physical activity definition can contribute to the text.

VERSION 2 – AUTHOR RESPONSE

Reviewer comment	What did we do	Where the changes are made(p-page/para-paragraph)
5. As a final observation, I consider that including the current definition of physical inactivity, together with the physical activity definition can contribute to the text.	Done as suggested	P7, para 1